# Reduced Tyrosine and Serine-632 Phosphorylation of Insulin Receptor Substrate-1 in the Gastrocnemius Muscle of Obese Zucker Rat

**Mohammad Shamsul Ola**

Department of Biochemistry, College of Science, King Saud University, Riyadh 11451, Saudi Arabia; mola@ksu.edu.sa; Tel.: +966-558013579

**Abstract:** Obesity has become a serious health problem in the world, with increased morbidity, mortality, and financial burden on patients and health-care providers. The skeletal muscle is the most extensive tissue, severely affected by a sedentary lifestyle, which leads to obesity and type 2 diabetes. Obesity disrupts insulin signaling in the skeletal muscle, resulting in decreased glucose disposal, a condition known as insulin resistance. Although there is a large body of evidence on obesity-induced insulin resistance in various skeletal muscles, the molecular mechanism of insulin resistance due to a disruption in insulin receptor signaling, specifically in the gastrocnemius skeletal muscle of obese Zucker rats (OZRs), is not fully understood. This study subjected OZRs to a glucose tolerance test (GTT) to analyze insulin sensitivity. In addition, immunoprecipitation and immunoblotting techniques were used to determine the expression and tyrosine phosphorylation of insulin receptor substrate-1 (IRS-1) and insulin receptor-β (IRβ), and the activation of serine-632-IRS-1 phosphorylation in the gastrocnemius muscle of Zucker rats. The results show that the GTT in the OZRs was impaired. There was a significant decrease in IRS-1 levels, but no change was observed in IRβ in the gastrocnemius muscle of OZRs, compared to Zucker leans. Obese rats had a higher ratio of tyrosine phosphorylation of IRS-1 and IRβ than lean rats. In obese rats, however, insulin was unable to induce tyrosine phosphorylation. Moreover, insulin increased the phosphorylation of serine 632-IRS-1 in the gastrocnemius muscle of lean rats. However, obese rats had a low basal level of serine-632-IRS-1 and insulin only mildly increased serine phosphorylation in obese rats, compared to those without insulin. Thus, we addressed the altered steps of the insulin receptor signal transduction in the gastrocnemius muscle of OZRs. These findings may contribute to a better understanding of human obesity and type 2 diabetes.

**Keywords:** obesity; diabetes; insulin signaling; insulin resistance; insulin receptor substrate





## 1. Introduction

Obesity has reached an epidemic level worldwide, with more than 650 million people reported to be obese in 2016, according to the World Health Organization [1]. The prevalence of obesity among people has significantly increased, mainly in developing and developed countries, due to people's sedentary lifestyle because of economic growth. The consumption of a high-glycemic-index diet and increased fat intake with a deficiency in physical activity lead to obesity, which ultimately initiates the progression of diabetes and its complications [2].

The skeletal muscle is the most extensive and essential tissue, where a significant part of glucose in the blood is taken up and metabolized within cells, facilitated by insulin, and aided by muscle contraction or exercise. Under normal physiological conditions, elevated blood glucose causes insulin release, activating the insulin receptor on the skeletal muscle cell membrane that phosphorylates the tyrosine residues of the insulin receptor substrate (IRS) associated with the insulin receptor (IR). Mainly, tyrosine-phosphorylated IRS-1 plays a significant and positive role in activating a series of downstream insulin

signaling that translocate glucose transporter-4 (GLUT4) to the cell membrane to transport a substantial part of the circulating glucose [3,4]. However, type 2 diabetes or obesity disrupts signaling mediated by IRS-1 in the skeletal muscle, by inhibiting the tyrosine phosphorylation of IRS-1 and/or activating the serine/threonine phosphorylation of IRS-1, thereby inactivating or downregulating insulin cascade that reduces glucose disposal, known as insulin resistance [3,5]. A large body of evidence is available on diabetes and obesity-induced insulin resistance in different skeletal muscles; however, the molecular mechanism of insulin resistance due to a disruption in insulin receptor signaling, specifically in the gastrocnemius skeletal muscle of obese Zucker rats, is not yet fully understood.

The reduced tyrosine phosphorylation and increased serine/threonine phosphorylation of IRS-1 due to diabetes or obesity have mostly been considered to be negative regulators of IRS-1 function and are widely regarded as critical players in inhibiting insulin signaling [6,7]. Paradoxically, several IRS-1 phosphorylation sites, including serine-307, serine-612, and serine-632, are stimulated by insulin, including with IRS-1 tyrosine phosphorylation [8–10]. However, several studies reported that the insulin-stimulated serine/threonine phosphorylation of IRS-1 can be increased, decreased, or remain unaffected [11]. Thus, these contradictory findings suggest a need for a more comprehensive study to understand the specific IRS-1 serine phosphorylation in insulin resistance under in vivo conditions. To characterize insulin resistance due to altered signaling through IRS-1, we employed obese Zucker rats (OZRs), a well-studied genetic model of insulin resistance that exhibits marked hyperinsulinemia due to a leptin receptor mutation [12,13]. In addition, KK-Ay/J heterozygous mice, which develop hyperglycemia, hyperinsulinemia, and glucose intolerance, are known to serve as excellent models of type 2 diabetes and were employed in a few experiments [14].

The OZRs are an excellent model for early-stage non-insulin-dependent diabetes mellitus (NIDDM) induced by overeating and being overweight. OZRs develop marked hyperglycemia and type 2 diabetes at 7–10 weeks [15,16]. The features of OZRs include insulin resistance, hyperinsulinemia, glucose intolerance, dyslipidemia, and compromised insulin signaling in the skeletal muscle and liver [17,18]. The primary focus of this investigation was to examine altered upstream intracellular factors, mainly signaling through the insulin receptor and insulin receptor substrate in the gastrocnemius muscle of OZRs that may cause insulin resistance. For this, we first examined the expression levels of IRS-1 and the activation of tyrosine phosphorylation of IRS-1 by insulin in the gastrocnemius muscle of OZRs, and a comparison was made with lean Zucker rats. Second, the expression levels of IRβ and the activation of tyrosine phosphorylation of IRβ with insulin were compared between obese Zucker and lean rats. Finally, we investigated alterations in the level of serine-632 phosphorylation of IRS-1, with and without insulin, in the gastrocnemius muscle of OZRs, and a comparison was made with lean rats. Based on our data analysis, we addressed the altered steps of the insulin receptor signal transduction level in the gastrocnemius muscle of OZRs. These findings may help develop a better understanding of human obesity and type 2 diabetes.

## 2. Materials and Methods

### 2.1. Animals

The Zucker rat colony was maintained in the animal facility at The Milton S. Hershey Medical Center, the Pennsylvania State University College of Medicine. Breeding pairs to initiate the colony were obtained from Marcelle Lavau, Inserm U177, Unite de Recherches sur la Physiolpathologie de la Nutrition, Paris, France. In the present experiments, obese female Zucker rats that were 8–10 weeks old and their age-matched control rats were used. Some studies used non-diabetic C57BL/6J controls or diabetic KK. Cg-Ay/J (KKay) was purchased from the Jackson Laboratory (Bar Harbor, ME, USA). KKay mice, 12–16 weeks old, and their aged-matched controls were used in the experiment. These rats/mice were kept on a 12 h light/dark cycle at constant room temperature, and a conventional laboratory diet and tap water were provided. The Milton S. Hershey Medical Center Institutional Animal

Care and Use Committee approved all procedures concerning these animals. These studies adhered to the National Institutes of Health Guide for the Care and Use of Laboratory Animals. There were 4–6 rats/mice per experimental group. Insulin (Humulin; Sigma Chemicals (St. Louis, MO, USA)) was intraperitoneally injected at a dose of 5 units/kg body weight into the rats. Five minutes after insulin injection, rats were euthanized, and gastrocnemius muscle was retracted and clamped at the Achilles tendon, freeze-clamped with tongs chilled in liquid nitrogen, and stored at $-80$ °C following the method of [19]. In the same way, the gastrocnemius muscle of mice was retracted and frozen in liquid nitrogen. Proteins were extracted from powdered, frozen tissue samples at 4 °C in the lysis buffer with a polytron homogenizer (Fisher, Pittsburgh, PA, USA). Animal experiments were conducted at Penn-State, Hershey, PA, USA; however, most of the molecular analyses of insulin signaling were carried out at King Saud University, KSA.

### 2.2. Glucose Tolerance Test (GTT)

Zucker lean (200 to 250 g body weight) and obese rats (300–375 g) fasted overnight. A glucose tolerance test was performed after an intraperitoneal injection of 1.5 g/kg of filtered glucose into each animal. Blood glucose level was measured in the tail blood samples before and after 15, 30, 60, and 120 min. Blood glucose levels were measured with a glucometer (Accu-check). Blood glucose concentrations were plotted as a function of time, and the areas under the curve were calculated. Overall changes in glucose during GTT were calculated as the area under the curve (AUC) and above the basal levels. The total area under the curve was calculated for glucose (AUC). The glucose index was used as a surrogate marker for insulin sensitivity.

### 2.3. Protein Extraction and Immunoprecipitation

Frozen powdered gastrocnemius tissues from OZRs and leans were transferred to Eppendorf tubes. Approximately 50–100 mg of muscle powder was given short sonication bursts 20 times in 50 mM HEPES buffer, pH 7.4, containing 137 mM NaCl, 1 mM $MgCl_2$, 1 mM $CaCl_2$, 2 mM $Na_3VO_4$, 10 mM sodium pyrophosphate, 10 mM NaF, 2 mM EDTA, 2 mM PMSF, 10 mM benzamidine, 10% glycerol, 1% NP-40, and a protease inhibitor tablet. Sample homogenates were kept on ice for 20 min and then centrifuged at $15,000\times g$ for 15 min. Supernatants were collected, and protein concentrations were estimated using the Bio-Rad DC protein assay kit (Bio-Rad). For immunoprecipitation, 2 mg of extracted protein in the supernatants from each group of rats was incubated overnight with monoclonal anti-IRS-1 or anti-IRβ antibodies (10 μg/mL) and then incubated with 20 μL of protein A-sepharose. Immune complexes were collected on protein A-sepharose beads (Pharmacia Fine Chemicals, Piscataway, NJ, USA), washed three times in RIPA buffer (0.15 M NaCl, 10 mM phosphate buffer, pH 7.0, 1% NP-40, 1% sodium deoxycholate, 0.1% sodium dodecyl sulfate), boiled in Laemmli's sample buffer for 5 min, and loaded onto 7.5–10% of SDS-PAGE gels for immunoblotting experiments.

### 2.4. Immunoblotting Analysis of IRS1, IRβ, and Tyrosine and Serine Phosphorylation Levels in the Gastrocnemius Muscle of Zucker Lean and Obese Rats

For immunoblotting of muscle-extracted protein, 50 μg of the total protein was boiled in a sample buffer and separated on 7.5% of SDS–PAGE gel and was then transferred to nitrocellulose (Promega, Madison, WI, USA). After transferring the separated proteins onto nitrocellulose membranes, the membranes were blocked for 1.5 h at room temperature with Tris-buffered saline–0.1% Tween-20 (TBS-T), containing 5% non-fat milk. After washing the membranes with TBS-T, they were incubated overnight with the polyclonal antibody against IRS-1 (Upstate Biotechnology, 1:1000 dilution, Lake Placid, NY, USA) or IRβ (Santa Cruz Biotechnology, Santa Cruz, CA, USA) based on published methods [20,21]. In addition, the immunoprecipitated membranes were incubated overnight using a monoclonal anti-pTyr (1:1000 dilution, Upstate Biotechnology, Inc.) and anti-phospho-ser632 (1:1000 dilution, Upstate Biotechnology). Membranes were washed 3–4 times with TBS-T and

then probed with anti-rabbit or anti-mouse horseradish peroxidase antibodies (Amersham) (1:5000 dilution) for 1.5 h and the proteins were visualized using the ECL Western detection system (SuperSignal West Pico (Pierce)). Following immunoblotting, films were scanned and bands were quantitated with a Bio-Rad Molecular Imager (GS-800). A drag-and-drop rectangular grid was placed on the band of interest and captured for processing, and the density was quantified by Quantity One Software (Bio-Rad). Background density was automatically subtracted.

### 2.5. Statistics

All values were expressed as means ± standard error of the mean (SEM). The significance between experimental groups was analyzed by two-way ANOVA with a post hoc Dunnett test using SPSS software (version 16; Chicago, IL, USA). The threshold of statistical significance was set at $p < 0.05$. Significant differences in figures are denoted as either * or # for lean versus obese rats.

## 3. Results

### 3.1. Impaired Glucose Tolerance Test (GTT) in the Obese Zucker Rats (OZRs)

At the time of the study, OZRs weighed significantly more than their lean littermates. As a result, basal blood glucose was significantly higher in the obese (7.0 mmol/L) than in the lean rats (5.3 mmol/L). The glucose tolerance test is a primary clinical parameter for diagnosing a diabetic state. Therefore, OZRs were subjected to a GTT. During glucose tolerance testing, the obese group had significantly higher blood glucose concentrations at all the time points (15, 30, 60, and 120 min; Figure 1A) after glucose administration. Furthermore, as the figure shows, the area under the curve (AUC) values, calculated using the GTT data, resulted in a significantly higher AUC value compared to the lean group, indicating impaired glucose tolerance in the OZRs (Figure 1B).

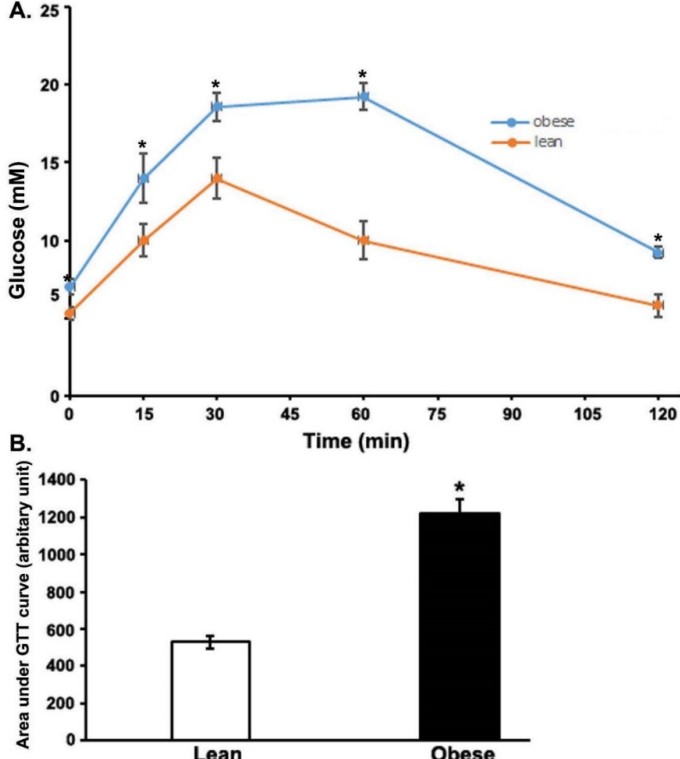

**Figure 1.** Glucose tolerance test (GTT). At time 0, rats were subjected to glucose (1.5 g/kg IP). (**A**). Blood glucose levels were determined at the indicated time points using a OneTouch Ultra glucometer (LifeScan, Milpitas, CA, USA) and (**B**). The area under the curve (AUC) was calculated using GraphPad PRISM software. * $p < 0.01$. In each group, six rats were used for GTT analyses.

### 3.2. Reduced IRS-1 Level in the Gastrocnemius Muscle of Obese Zucker Rats and Non-Obese Insulin Resistance in KKay Mice

To determine the muscle IRS-1 expression levels, we performed immunoblotting to measure the amounts of IRS-1 protein in the muscle of obese and lean Zucker rats and in the gastrocnemius muscle of insulin-resistant KKay and control mice (Figure 2). Densitometry analyses of the protein bands indicated that IRS-1 protein levels were significantly decreased to almost 55% in the muscle of OZRs, compared to the lean rats ($p < 0.01$). Furthermore, similar to OZRs, in the gastrocnemius muscle of KKay mice, the IRS1 levels decreased to almost 65%, compared to the control levels ($p < 0.01$) (Figure 2D,E). However, the protein expression of IRβ indicated no significant change between lean rats and OZRs.

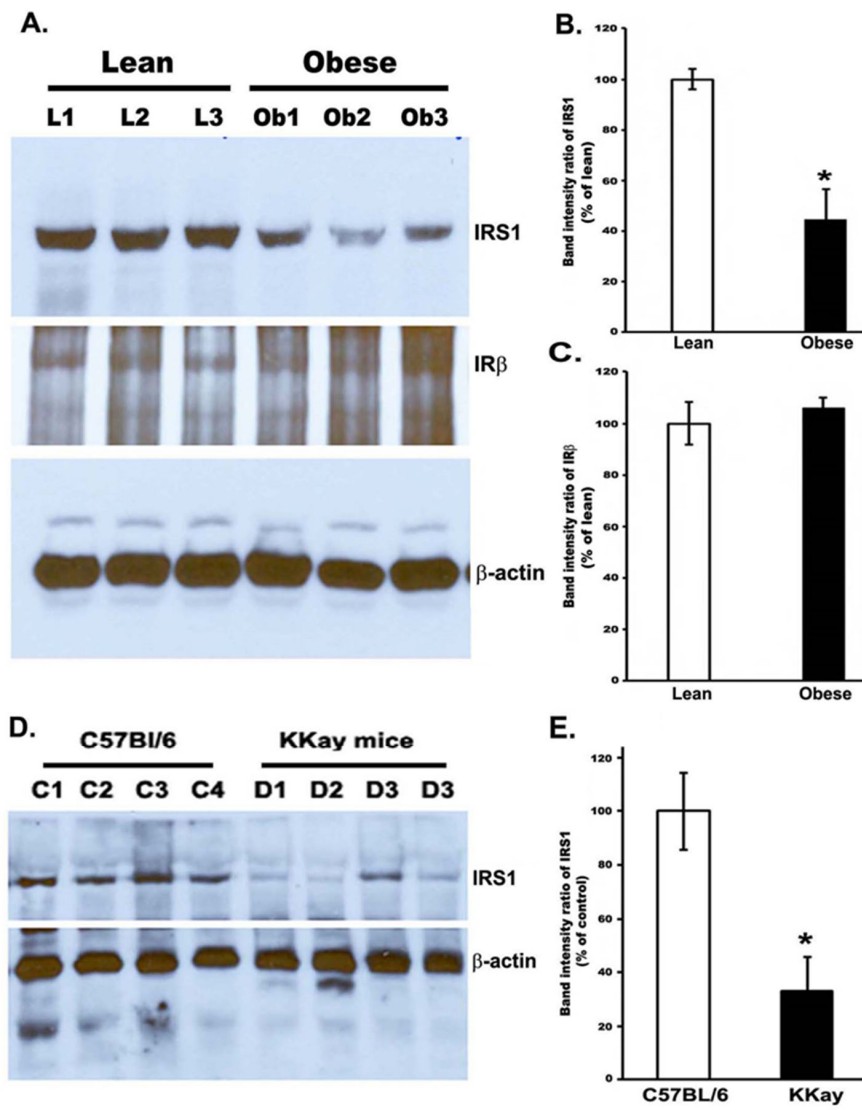

**Figure 2.** Expression of IRS-1 and IRβ protein in the gastrocnemius muscle of Zucker rats and KKay mice. Protein was isolated from the muscle as described in the Methods section. (**A**). The amount of IRS-1 protein was determined by immunoblotting with anti-IRS-1 and anti-IRβ antibodies (N = 6). Band intensity was quantitated with a Bio-Rad Molecular Imager. (**B**). Band intensity of IRS1 in obese rats in terms of % of control (lean rats). (**C**). Band intensity of IRβ in terms of % of control (lean rats). (**D**). Expression of IRS1 in the gastrocnemius muscle of KKay and C57BL/6 mice (N = 4). (**E**). Band intensity of IRS1 in KKay mice in terms of % of control (non-diabetic C57BL/6 mice). * $p < 0.01$; vs. Zucker lean rats.

### 3.3. Altered Insulin-Stimulated Phosphotyrosine of IRβ (pY-IRβ) in the Gastrocnemius Muscle of Obese Zucker Rats

To evaluate the ability of insulin to activate the tyrosine phosphorylation of the IRβ, we immunoprecipitated insulin receptors from protein extracts prepared from the gastrocnemius muscle of insulin-stimulated lean and obese rats immunoblotted with anti-phosphotyrosine antibody (anti-pY). Densitometric analyses of the bands indicated that the tyrosine phosphorylation of insulin receptors (pY-IRβ) increased at least four-fold in the insulin-stimulated lean rats, compared to unstimulated leans. In the case of obese rats, the hyperphosphorylation of tyrosine residues of IRβ was observed (~two-fold increase) in the muscle compared to lean rats (Figure 3). However, insulin stimulation in obese rats did not significantly affect the pY-IRβ level, as compared to unstimulated obese rats. The basal hyper-tyrosine phosphorylation of IRβ and the non-stimulation of pY- IRβ after insulin injection into the muscle of OZRs established the disruption of insulin signaling, causing insulin resistance. This might be due to a disturbance in the tyrosine kinase activity of the insulin receptor in OZRs.

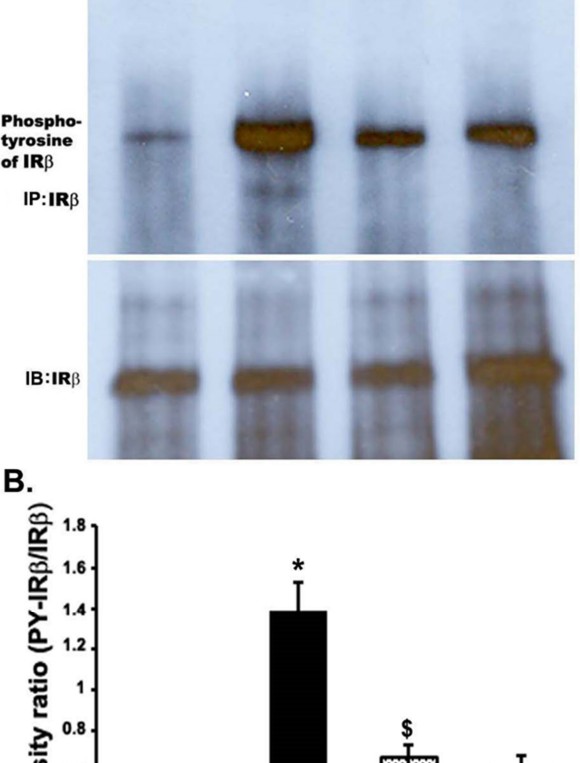

**Figure 3.** Tyrosine phosphorylation of IRβ in the gastrocnemius muscle of Zucker rats. Rats were anesthetized and 5 units of insulin/kg were intraperitoneally injected into both lean and obese rats. Then, 5 min later, gastrocnemius skeletal muscle was excised and homogenized in an extraction buffer at 4 °C, as described in the Methods section. After centrifugation, aliquots with the same amount of protein were immunoprecipitated with anti-IRβ antibodies at 4 °C and, subsequently, protein A-sepharose. (**A**). Immunoprecipitated proteins were immunoblotted with anti-pY-IRβ antibody. (**B**). Bands were quantitated with a Bio-Rad Molecular Imager and plotted as band intensity ratio. * $p < 0.01$; $^{\$}$ $p < 0.05$: both vs. corresponding lean value. In each group, at least four rats were used.

### 3.4. Altered Insulin-Stimulated Tyrosine Phosphorylation of IRS-1 (pY-IRS-1) in the Gastrocnemius Muscle of Obese Zucker Rats

To assess the tyrosine phosphorylation of IRS-1 by insulin in the gastrocnemius muscle of both lean rats and OZRs, muscle homogenates were subjected to immunoprecipitation with anti–IRS-1 antibody and analyzed by immunoblotting with a specific anti-phosphotyrosine antibody. There was a low basal level of the tyrosine phosphorylation of IRS-1 (pY-IRS-1) in the gastrocnemius muscle without insulin stimulation in the lean rats (Figure 4). However, insulin stimulation resulted in an almost three-fold increase in pY-IRS-1 in the muscle of the lean control rats (lean+Ins). Furthermore, when we compared the data in terms of the ratio of pY-IRS-1 with the total amount of IRS-1 in those insulin-stimulated and -unstimulated lean and obese samples, to our surprise, pY-IRS-1 was significantly increased in the obese rats, compared to both lean and lean-stimulated rats with insulin (Figure 4). This was mainly due to decreased IRS-1 protein in obese rats. However, in the case of insulin-stimulated obese rats, the pY-IRS-1 level was significantly reduced compared to insulin-stimulated lean and unstimulated obese rats.

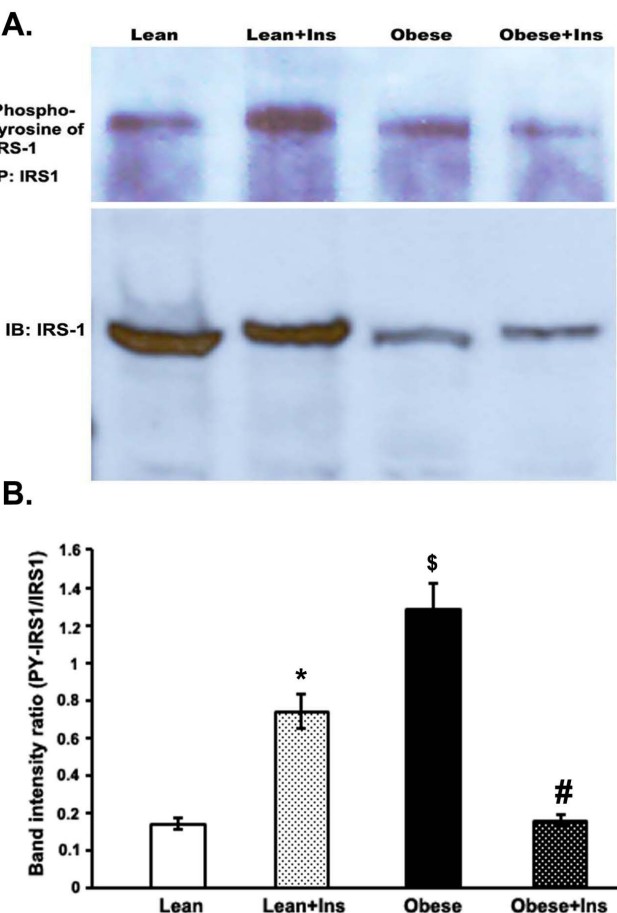

**Figure 4.** Tyrosine phosphorylation of IRS-1 in the gastrocnemius muscle of Zucker rats. Rats were anesthetized, and 5 units of insulin were intraperitoneally injected into both lean and obese rats. Then, 5 min later, gastrocnemius skeletal muscle was excised and homogenized in an extraction buffer at 4 °C, as described in the Methods section. After centrifugation, aliquots with the same amount of protein were immunoprecipitated with anti-IRS-1 antibodies at 4 °C and, subsequently, protein A-sepharose. (**A**). Immunoprecipitated proteins were immunoblotted with anti-pY-IRS-1 antibody. (**B**). Bands were quantitated with a Bio-Rad Molecular Imager and plotted as band intensity ratios of pY-IRS-1 to IRS-1. * $p < 0.01$; $ $p < 0.01$ vs. corresponding lean value; # $p < 0.05$ vs. lean + insulin. In each group, at least four rats were employed.

### 3.5. Insulin-Stimulated Decrease in Serine Phosphorylation of IRS-1 (Serine 632-IRS-1) in the Gastrocnemius Muscle of Obese Zucker Rats

To determine the activation of the serine phosphorylation of IRS-1 by insulin in the gastrocnemius muscle of both lean rats and OZRs, muscle homogenates were subjected to immunoprecipitation with anti-IRS-1 antibody and analyzed by immunoblotting with a specific anti-serine-632 IRS-1 antibody. The serine-632 residue was located close to one of the tyrosine-phosphorylated motifs of IRS-1 associated with PI 3-kinase. Using a specific anti-phospho-Ser632 antibody, we found that the basal serine 632 phosphorylation of IRS-1 was three-fold lower in the muscle of OZRs than in lean rats ($p < 0.005$; Figure 5). The amount of serine-632 IRS-1 phosphorylation increased at least five-fold more in the muscle of insulin-stimulated lean rats than in unstimulated lean rats (Figure 5B). However, the insulin-stimulated phosphorylation level of serine-632-IRS-1 in obese muscle was significantly lower than insulin-stimulated lean rats (Figure 5B).

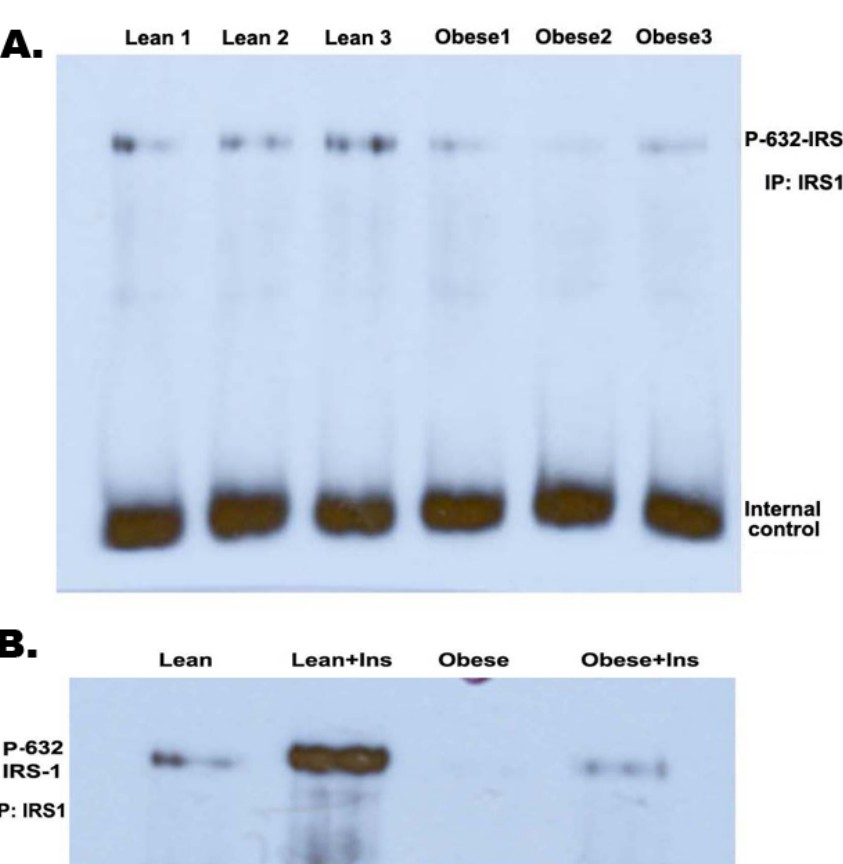

**Figure 5.** *Cont.*

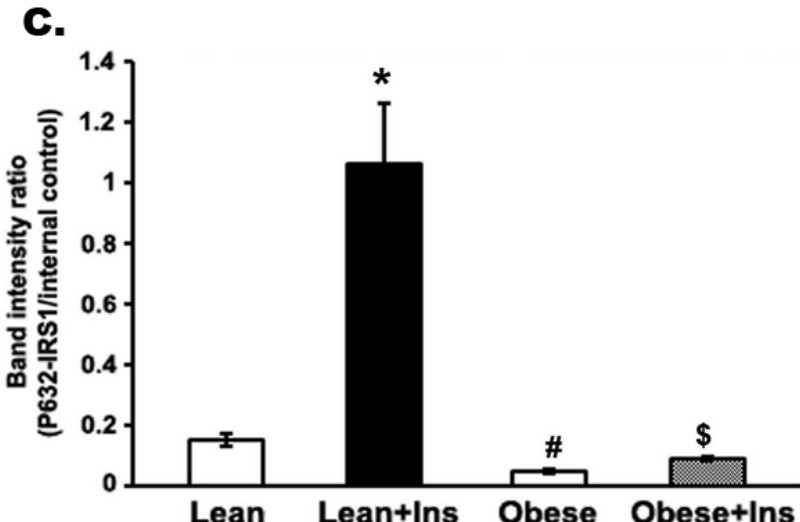

**Figure 5.** Serine-632 phosphorylation of IRS-1 in the gastrocnemius muscle of Zucker rats. Rats were anesthetized, and 5 units of insulin were intraperitoneally injected into both lean and obese rats. Then, 5 min later, gastrocnemius skeletal muscle was excised and homogenized in an extraction buffer at 4 °C, as described in the Methods section. After centrifugation, aliquots with the same amount of protein were immunoprecipitated with anti-IRS1 antibodies at 4 °C and, subsequently, protein A-sepharose. Immunoprecipitated proteins were immunoblotted with anti-serine-632-IRS-1 antibody, and bands were quantitated with a Bio-Rad Molecular Imager. (**A**). Representative immunoblot of serine-632-IRS-1 phosphorylation in lean rats and OZRs. (**B**). Immunoblot of serine-632-IRS-1 in lean rats and OZRs with and without insulin stimulation. (**C**). Bar graph showing band intensity ratios of serine-632-IRS1 phosphorylation in terms of internal control (IgG ab). * $p < 0.01$; # $p < 0.05$ vs. corresponding lean value; $ $p < 0.01$ vs. lean + insulin. In each group, five rats were employed.

## 4. Discussion

Obesity is a metabolic syndrome associated with several comorbidities, including type 2 diabetes and its complications. Obesity-induced altered metabolites, including free fatty acids and proinflammatory cytokines, have been considered major factors connected with insulin resistance, the characteristic of type 2 diabetes [3,22,23]. In this study, OZRs served as a model of insulin resistance to understand the altered upstream insulin signaling site, especially at the level of insulin receptors and its substrate in the gastrocnemius muscle. The skeletal muscle is insulin stimulation's primary site of glucose disposal [24]. We determined altered levels of IRS-1, tyrosine phosphorylation, and serine-632 phosphorylation of IRS-1 in the gastrocnemius muscle of OZRs, mostly considered to be a marker of insulin resistance for obesity and type 2 diabetes.

We investigated an impaired GTT and a significantly low amount of IRS-1 in the gastrocnemius muscle of both OZRs (~55%) and insulin-resistant KKay mice (~65%), the latter being a typical type 2 diabetic model, compared to their controls [25]. These results correlate with previous reports on the hind limb and quadriceps muscle of OZRs [26–28] and the muscle of obese diabetic mice [29], which are well supported by human studies in both type 2 diabetic and obese subjects [30–32]. The precise mechanism of the decreased level of IRS1-1 protein in obesity and type 2 diabetes is unknown; however, serine/threonine phosphorylation has been proposed as the potential mechanism in accelerated degradation [33]. Thus, a lowered IRS-1 protein alone in the muscle of OZRs might trigger the progression of insulin resistance since this defect would diminish downstream insulin signaling to cause a reduction in glucose disposal. In contrast, other studies could not demonstrate a deficiency in IRS-1 protein expression in the muscle cells of type 2 diabetes and several other tissues of obese animals [34–36]. Therefore, the discrepancy in the protein expression of IRS-1 might be due to the differences in the types of muscles used.

Furthermore, as opposed to some previous studies, we did not find a substantial alteration in the expression level of IRβ protein in the gastrocnemius muscle of OZRs, when compared to leans. Many studies reported a decreased amount of insulin receptors in muscle and other tissues of obese animals, obese and hypertensive rats, and high-fat-fed-induced insulin-resistant rats and mice [36–40]. Our data indicate that insulin receptor levels in gastrocnemius muscle alone might not directly cause insulin resistance; somewhat altered phosphorylation or affected downstream signaling may account for a decreased insulin sensitivity in OZRs. Normally, insulin activates insulin receptor tyrosine kinase to autophosphorylate the tyrosine residues, which leads to the phosphorylation of intracellular receptor substrates transmitting downstream signaling for glucose transport. However, altered levels of tyrosine phosphorylation of both the insulin receptor and its substrate have been found in the muscle of various obese and Type 2 diabetic models [31,36]. In this study, as expected, insulin activated the pY-IRβ level to several folds in the muscle of the lean rats, compared to those without insulin. However, obese rats, being insulin-resistant, had significantly lower levels of insulin-stimulated pY-IRβ compared to their leans, which is consistent with previous studies [31,36,41]. Surprisingly, a significant increase in the basal level of pY-IRβ in obese rats was observed compared to lean rats. However, insulin did not cause the further activation of pY-IRβ in the obese rats, compared to those without insulin after correction for IRβ proteins. This confirms the hyperinsulinemic condition in OZRs, which may initiate insulin resistance at the level of altered tyrosine phosphorylation of IRβ.

Furthermore, unlike several previous studies, no apparent variation in the basal level of pY-IRS1 in the gastrocnemius muscle of OZRs was observed compared to their lean counterparts [34,40,42]. However, based on reductions in the level of IRS-1 proteins in the muscle of obese rats and after correction of the protein, the ratio of the basal level of pY-IRS-1 to IRS-1 increased, compared to lean rats. Moreover, the ratio of insulin-activated pY-IRS1 to IRS-1 decreased in the muscle of OZRs, compared to obese and insulin-activated lean rats, mainly due to a reduced level of IRS-1 and partly due to a decreased pY-IRβ in the insulin-activated OZRs. Previous studies have demonstrated that the resistance to insulin stimulation is due to impaired tyrosine kinase activity in the muscle of obese rats that reduces the tyrosine phosphorylation of insulin receptors and their substrate [36,43,44]. However, a few studies reported no change in pY-IRS-1 in the muscle of Ob/Ob mice and obese rats with diabetes, even after insulin activation [45]. As Anai et al. (1998) suggested, the discrepancies in insulin-stimulated IRS-1 phosphorylation among animal models could be partly due to the difference in blood glucose levels [27].

Another primary reason for insulin resistance is an altered serine/threonine phosphorylation of IRS-1 by a large number of protein kinases, a critical event explaining the molecular basis of the down-regulation of insulin signaling. Many serine/threonine kinases phosphorylate at different serine/threonine residues of IRS-1 to cause inhibition of insulin signaling by negating IRS-1's ability to aid as a mediator of insulin receptor tyrosine kinase signals [46]. However, several studies reported on the positive and negative regulatory roles pertaining to the serine/threonine phosphorylation of the insulin receptor substrate [11,47,48]. Our results on the serine phosphorylation of IRS-1 in the muscle of lean rats indicate that insulin-stimulated serine-632 IRS-1 phosphorylates to several folds compared to without insulin. This is consistent with the study of [9,35], showing that insulin treatment promoted serine-632 IRS-1 phosphorylation in the skeletal muscle of mice. As Bouzakri et al. (2003) suggested, the activation of the MAP kinase in response to insulin in type 2 diabetic muscle cells may lead to increased serine-636, equivalent to rodent's serine-632 IRS-1 phosphorylation [35]. However, we found diminished basal serine-632 IRS-1 phosphorylation in obese muscle, and insulin only mildly activated serine phosphorylation, compared to unstimulated obese, as opposed to previous studies showing the hyperphosphorylation of serine-632 IRS-1 in the muscle cells of type 2 diabetes [35,49,50]. In hyperinsulinemia, such as in obesity or type 2 diabetes, the decreased serine-632 IRS-1 phosphorylation could exacerbate insulin resistance. Indeed, a study reported that replacing IRS-1 serine-632 with alanine caused a significant inhibition of

insulin-stimulated IRS-1 phosphorylation, while another study suggested that mutation of serine 632-IRS-1 decreased insulin signaling [51,52]. Thus, impaired IRS-1 level and tyrosine and/or serine-632 IRS-1 phosphorylation in the gastrocnemius muscle of OZRs may be an important mechanism contributing to the pathogenesis of insulin resistance in obesity and type 2 diabetes.

**Funding:** This research was funded by King Abdul Aziz City for Science and Technology (KACST-NPST), grant number 13-MED-1374, at the Department of Biochemistry, College of Science, King Saud University, Riyadh, KSA.

**Institutional Review Board Statement:** The Milton S. Hershey Medical Center Institutional Animal Care and Use Committee approved all procedures concerning these animals. These studies adhered to the National Institutes of Health Guide for the Care and Use of Laboratory Animals.

**Informed Consent Statement:** Not applicable.

**Data Availability Statement:** All relevant data are included within the manuscript. The raw data supporting the findings of this manuscript will be provided by the author to any researcher on reasonable request.

**Acknowledgments:** The author would like to thank Kathryn F. LaNoue, at the College of Medicine, Penn-State University, Hershey, USA, for her support in providing facilities for animal experiments. The author also extends their appreciation to the financial support from KACST-NPST (13-MED-1374).

**Conflicts of Interest:** The author declares no conflict of interest. The funders had no role in the design of the study; in the collection, analyses, or interpretation of data; in the writing of the manuscript; or in the decision to publish the results.

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
