# Peer review of "Reduced Tyrosine and Serine-632 Phosphorylation of Insulin Receptor Substrate-1 in the Gastrocnemius Muscle of Obese Zucker Rat"

_cimb, doi:10.3390/cimb44120410_

Round 1

Reviewer 1 Report

This study investigates insulin signaling in the gastrocnemius muscle of obese Zucker rats and KKay mice. The author found that glucose tolerance was impaired in obese Zucker rats. The protein levels of IRS1 were significantly lower in the gastrocnemius muscle of obese Zucker rats and KKay mice, compared to their control animals. Also, Zucker rats had impaired insulin-stimulated tyrosine phosphorylation of IRS-1 and IRβ as well as Serine-632 phosphorylation of IRS-1 in the gastrocnemius muscle. Based on these data, the author concluded that altered steps at the insulin receptor signal transduction were addressed in the gastrocnemius muscle of obese Zucker rats.

As the author noted, the results presented in this manuscript are similar to and are supported by several previous studies reporting that insulin signaling is impaired in the muscle of obese Zucker rats. While I am sure that when the results of this study are added to the existing literature, the results will add another new example yet may have limited impact on the field since the findings are largely descriptive and confirmative. The author may want to clearly describe and emphasis on what are the novel aspects of the findings described in this manuscript.

Minors:

-        Figure 1A: the t-test should not be used here. Performing a two-way ANOVA with post hoc tests would be more appropriate over running t-tests.

-        It would be helpful for readers to explain what KKay mice are in the Introduction.

-        Please indicate the number of samples, i.e. the number of mice/rats used in each figure.

Author Response

Replies to reviewers’ comments:

Reviewer 1: Comments and Suggestions for Authors

This study investigates insulin signaling in the gastrocnemius muscle of obese Zucker rats and KKay mice. The author found that glucose tolerance was impaired in obese Zucker rats. The protein levels of IRS1 were significantly lower in the gastrocnemius muscle of obese Zucker rats and KKay mice, compared to their control animals. Also, Zucker rats had impaired insulin-stimulated tyrosine phosphorylation of IRS-1 and IRβ as well as Serine-632 phosphorylation of IRS-1 in the gastrocnemius muscle. Based on these data, the author concluded that altered steps at the insulin receptor signal transduction were addressed in the gastrocnemius muscle of obese Zucker rats.

As the author noted, the results presented in this manuscript are similar to and are supported by several previous studies reporting that insulin signaling is impaired in the muscle of obese Zucker rats. While I am sure that when the results of this study are added to the existing literature, the results will add another new example yet may have limited impact on the field since the findings are largely descriptive and confirmative. The author may want to clearly describe and emphasis on what are the novel aspects of the findings described in this manuscript.

Reply: First, I would like to thank the reviewer for reading and making positive comments. The reviewer raised his/her concern about the basis/justification of the study and also the novel aspects of the findings. As indicated in the last sentence of the second paragraph and third paragraph of the introduction section, based on the available literature, evidence is available on diabetes and obesity-induced insulin resistance in the different skeletal muscles; however, the molecular mechanism of insulin resistance due to disruption in insulin receptor signaling, specifically in the gastrocnemius skeletal muscle of obese Zucker rats, was not yet fully understood. In addition, literature suggested contradictory findings in insulin-stimulated serine/threonine phosphorylation of IRS1. This clearly indicated a strong reason, why I proceeded to carry out the study using obese Zucker rats.

Novel aspects of the study

  1. 1. Previously, Insulin signaling was studied in the hind limb, quadriceps, and skeletal muscle but not in the gastrocnemius muscle of Obese Zucker rats, especially at the level of insulin receptor based on literature search until now.
  2. 2. IRS1 deficiency was found in the gastrocnemius muscle of obese rats, however, several studies reported no change, as discussed in the discussion section, which may be due to differences in muscle types (References 33-25; 2nd paragraph discussion section).
  3. 3. I found no change in IRexpression but others reported a decreased level (ref 35-39; 3rd paragraph discussion section).
  4. 4. Surprisingly, a significant increase in pY-IRin the gastrocnemius muscle of Obese rats (3rd paragraph discussion section).
  5. 5. Unlike several previous reports, no apparent variation in the basal level of pY-IRS1 in the gastrocnemius muscle of obese rats compared to leans (4th paragraph discussion section).
  6. 6. The ratios of the basal level of pY-IRS1 to IRS1 increased in obese gastrocnemius rats compared to lean. Mostly, the literature is not reported in terms of ratio (pY-IRS1/IRS1), which may give a different conclusion due to a decreased level of IRS1 due to diabetes. Even if pY-IRS1 is slightly decreased or there is no change in the obese muscle, after correction of the IRS1, the basal level of pY-IRS1 to IRS1 may be increased, as discussed in the (4th paragraph discussion section).
  7. 7. In the literature, it is indicated serine/threonine phosphorylation of IRS1 can have positive and negative regulatory roles in insulin signaling. Here I found a positive regulatory role of serine 632 IRS1 phosphorylation in the gastrocnemius muscle of Zucker rats. However, in the obese Zucker rats, diminished basal serine 632 IRS1 phosphorylation was observed (5th paragraph discussion section)

As summarized the main points of our results above and discussed in the discussion section, clearly, these results and findings are not similar, as the reviewer pointed out, but rather indicate the novel aspects of the study.

Minors:

-        Figure 1A: the t-test should not be used here. Performing a two-way ANOVA with post hoc tests would be more appropriate over running t-tests.

 Reply: I agree with the reviewer; in this edited manuscript, I have inserted a two-way ANOVA for comparisons between groups.

-        It would be helpful for readers to explain what KKay mice are in the Introduction.

 Reply: Yes, I have inserted about KKay mice in the introduction section.

-        Please indicate the number of samples, i.e. the number of mice/rats used in each figure.

Reply: Thanks; In the legend, I have inserted the number of mice/rats used in each figure in terms of “n”.

Reviewer 2 Report

Were the studies performed in male and female mice?

Please justify why it was not performed in male rats ?

For figure 3,4, 5 a two way ANOVA has to be performed

not sure how figure 4 was quantified. 

Author Response

Reviewer 2: Comments and Suggestions for Authors

Were the studies performed in male and female mice? Please justify why it was not performed in male rats ?

Reply: First, I would like to thank the reviewer for making positive comments. Our previous laboratory (Prof. Kathryn LaNoue and the group) has published extensively using female obese Zucker rats. In one of their publications, they reported no gender-related differences except in body weight and percentage of body fat (Xu B et al Am J Physiol 1998 Feb;274(2):E271-9). Studies using obese female Zucker rats make a better correlation with an obese human female since several reports indicated obesity is more prevalent in women than men in most countries (Cooper AJ Current Obes Report 2021 Dec;10(4):458-466)

For figure 3,4, 5 a two way ANOVA has to be performed not sure how figure 4 was quantified. 

Reply: Thanks, Yes, for these figures, two-way ANOVA is most appropriate for statistical analyses. I have edited and inserted it in the method section.

Method of quantification for figure 4: This experiment is based on immunoprecipitation of IRS1 from both lean and obese groups. Since IRS1 is decreased in the obese group, it is speculated that the tyrosine phosphorylation level of IRS1 might be low due to decreased IRS1, but this might not necessarily correlate with the IRS1 level. It is possible that less IRS1 might get high tyrosine phosphorylation under different conditions. For this reason, if we compare the ratio (pY-IRS1 to IRS1) between lean and obese groups, this would better estimate the extent of tyrosine phosphorylation even if IRS1 protein is low in obese or when treated with insulin. A comparison between those groups in terms of ratios (pY-IRS1/IRS1) would correct for the IRS1 protein and would give a better quantification of tyrosine phosphorylation of IRS1, which is the key element of positive insulin signaling.

Round 2

Reviewer 2 Report

In the methods it is written that 1.5g is given of glucose, but in the figure it is written as 1.25

After 2way ANOVA, is there an interaction? If there is no interaction , posthoc cannot be performed. Then main effects have to be showed. 

There is only one author in the manuscript? Were the molecular experiments performed in Penn or at KSA? Where tissues shipped to KSA? Please include in methods. Is the contribution of Kathryn F. LaNoue acknowledged. In other words, is Dr. LaNoue or the study team at Penn State OK with no other authors from the group. Please address in comments to editor. 

Author Response

Comments and Suggestions for Authors

Thanks to the reviewer for making constructive criticism and positive comments.

In the methods it is written that 1.5g is given of glucose, but in the figure it is written as 1.25

Reply: Apology for the mistake, I have corrected the same 1.5g in both places.

After 2way ANOVA, is there an interaction? If there is no interaction , posthoc cannot be performed. Then main effects have to be showed. 

Reply: Yes, I did two-way ANOVA with Posthoc, since there was interaction among the group. The main effects, and the significant “P” values were presented in the results and figure legends section.

There is only one author in the manuscript? Were the molecular experiments performed in Penn or at KSA? Where tissues shipped to KSA? Please include in methods. Is the contribution of Kathryn F. LaNoue acknowledged. In other words, is Dr. LaNoue or the study team at Penn State OK with no other authors from the group. Please address in comments to editor. 

Reply: I wrote in detail in my previous revision and explained this question of single authorship to the editor. I have acknowledged the late Dr.LaNoue in the acknowledgment section. In this study, I did not get help from anyone at King Saud University or Penn-State. I have added a sentence in the method section about the location of experimental analyses.

Part of my earlier reply to the editor about the single authorship:

This work was initiated when I was still in Penn-State University but I could not finish it mainly due to the reason I moved to King Saud University for a better job opportunity. But a few times, during my summer vacation, I got a chance to go back to accomplish some of the research work over there and also could bring some samples with me to do some molecular analyses here in my lab at King Saud University. This work was solely my idea, execution, and also experimental work, data analyses, and writing; everything was done by me. This is why it took a long time but I finally could accomplish it to be submitted for publication. In fact, I wanted some more studies related to this work but the facilities here did not allow me to proceed, and also I can’t wait any longer to publish this work. As evident from my acknowledgment in the manuscript, I acknowledged facilities and animal studies at Penn-state University with the help and support of the late Prof. Kathryn LaNoue.
